# Hardening of Respiratory Syncytial Virus Inclusion Bodies by Cyclopamine Proceeds through Perturbation of the Interactions of the M2-1 Protein with RNA and the P Protein

**DOI:** 10.3390/ijms241813862

**Published:** 2023-09-08

**Authors:** Cédric Diot, Charles-Adrien Richard, Jennifer Risso-Ballester, Davy Martin, Jenna Fix, Jean-François Eléouët, Christina Sizun, Marie-Anne Rameix-Welti, Marie Galloux

**Affiliations:** 1Institut Pasteur, Université Paris Cité, M3P, F-75015 Paris, France; cedric.diot@pasteur.fr; 2INSERM, UMR 1173 (2I), Université Paris-Saclay-Versailles St. Quentin, M3P, F-78180 Versailles, France; jennifer.risso-ballester@uvsq.fr; 3INRAE, Unité de Virologie et Immunologie Moléculaires (VIM), Université Paris-Saclay-Versailles St. Quentin, F-78350 Jouy-en-Josas, France; charles-adrien.richard@inrae.fr (C.-A.R.); davy.martin@inrae.fr (D.M.); jenna.fix@inrae.fr (J.F.); jean-francois.eleouet@inrae.fr (J.-F.E.); 4Institut de Chimie des Substances Naturelles, CNRS, Université Paris-Saclay, F-91190 Gif-sur-Yvette, France; christina.sizun@cnrs.fr; 5Laboratoire de Microbiologie, Hôpital Ambroise Paré, Assistance Publique des Hôpitaux de Paris, DMU15, F-75015 Paris, France

**Keywords:** RSV, cyclopamine, M2-1-P interaction, antiviral mechanism, inclusion bodies

## Abstract

Respiratory syncytial virus (RSV) RNA synthesis takes place in cytoplasmic viral factories also called inclusion bodies (IBs), which are membrane-less organelles concentrating the viral RNA polymerase complex. The assembly of IBs is driven by liquid-liquid phase separation promoted by interactions between the viral nucleoprotein N and the phosphoprotein P. We recently demonstrated that cyclopamine (CPM) inhibits RSV multiplication by disorganizing and hardening IBs. Although a single mutation in the viral transcription factor M2-1 induced resistance to CPM, the mechanism of action of CPM still remains to be characterized. Here, using FRAP experiments on reconstituted pseudo-IBs both in cellula and in vitro, we first demonstrated that CPM activity depends on the presence of M2-1 together with N and P. We showed that CPM impairs the competition between P and RNA binding to M2-1. As mutations on both P and M2-1 induced resistance against CPM activity, we suggest that CPM may affect the dynamics of the M2-1-P interaction, thereby affecting the relative mobility of the proteins contained in RSV IBs. Overall, our results reveal that stabilizing viral protein-protein interactions is an attractive new antiviral approach. They pave the way for the rational chemical optimization of new specific anti-RSV molecules.

## 1. Introduction

The replication of many viruses depends on the formation during infection of cellular biomolecular condensates through a liquid-liquid phase separation (LLPS) mechanism driven by low-affinity interactions of multivalent proteins [1,2,3]. Viral RNA synthesis of human and bovine respiratory syncytial viruses (hRSV and bRSV, respectively) has been shown to occur in cytoplasmic membrane-less viral factories, also termed inclusion bodies (IBs) [4,5]. RSV belongs to the *Mononegavirales* order (MNVs), which gathers single-stranded negative sense RNA viruses [6]. The genomic RNA is tightly encapsidated by the viral nucleoprotein N, forming helical ribonucleocapsids (NCs), which are the template for both the viral transcription and replication. RSV IBs concentrate all the elements required for viral RNA synthesis, i.e., the encapsidated genomic and antigenomic viral RNA, the viral polymerase L and its co-factor P, and the viral transcription factor M2-1. The N and P proteins are sufficient to induce LLPS and pseudo-IBs morphogenesis in cells and condensation in vitro [7,8]. Furthermore, RSV IBs contain subcompartments called IBAGs (IBs associated granules), where viral mRNAs and the viral transcription factor M2-1 specifically concentrate [4].

hRSV is the main cause of severe lower respiratory tract infections in children worldwide, and the leading cause of hospitalization of children under 6 months of age [9]. Re-infections occur throughout life, leading to severe lower respiratory infections in immunocompromised and elderly people, with a burden comparable to influenza virus [10]. Unfortunately, there is still no curative antiviral drug available to treat RSV infection. Similarly, bRSV is a major cause of respiratory disease in young calves [11], and commercial bRSV vaccines remain poorly efficient to protect animals [12,13].

We have previously shown that cyclopamine (CPM), a plant alkaloid known to target the Hedgehog pathway, inhibits hRSV replication both in cell culture and in vivo in a mouse model [14,15]. We have also recently shown that CPM inhibits bRSV replication [16]. CPM acts by hardening hRSV IBs and is suspected to target M2-1 protein, as the R151K mutation in M2-1 induces resistance to CPM treatment. However, no direct interaction between M2-1 and CPM could be observed. Thus, although CPM represents a new potential antiviral against RSV, its mechanism of action needs to be characterized to consider further improvements of the molecule.

The recruitment of M2-1 into IBs depends on its interaction with P, and the M2-1-P interaction is required for the dephosphorylation of M2-1 by the cellular phosphatase PP1, also recruited by the P protein [17,18]. Dephosphorylation of M2-1 induces a switch of interaction from P to mRNA, leading to the co-localization of M2-1 with viral mRNA in IBAGs. RSV M2-1 is a 22 kDa protein that forms tetramers in solution [19,20]. It is composed of an N-terminal zinc-finger domain, a central oligomerization domain, and a C-terminal globular domain (M2-1 core). hRSV M2-1 displays a compact disk-shaped tetramer, referred to as a closed conformation [21]. However, it has been suggested that M2-1 might display conformational plasticity, by analogy with the M2-1 protein of the closely related human metapneumovirus (HMPV), where the core domain of one M2-1 protomer is projected to the outside of the disk in an open conformation [22]. The hRSV M2-1 core domain interacts with both the P protein and viral mRNA, in a competitive manner [19]. Binding surfaces of P and RNA on M2-1 core domain partly overlap [23]. The M2-1 binding motif of P was shown to encompass residues 95–110 of P [24]. Furthermore, crystallographic data of hRSV M2-1 in complex with short RNA disclosed a dual RNA-binding mode [25]: the M2-1 zinc finger domain contains an RNA binding site that recognizes an RNA base, whereas the M2-1 core domain binds to the RNA backbone, thus with no sequence specificity expected [23]. Moreover, M2-1 displays annealing of longer RNA with secondary structures, resulting in cooperative binding to two sites with increased affinity [26]. These data highlight the complexity of M2-1 interactions with both P and RNA. Of note, the M2-1 R151 residue, which induces resistance to CPM when mutated into a lysine, is critical for the interaction with RNA, and also involved in P-binding [21,23]. However, previous data suggest that CPM does not impair the interaction of M2-1 with RNA, nor that with P [14].

In the present study, we aimed at shedding light on the mechanism of action of CPM. Using fluorescence recovery after photobleaching (FRAP) and hypotonic shock experiments, we first showed that CPM induces hardening of pseudo-IBs formed in cells upon the transient expression of hRSV N and P only when M2-1 is co-expressed. Similar results were obtained in vitro by co-incubation of recombinant N, P, and M2-1 proteins. The study of M2-1 interaction with P and RNA using in vitro assays revealed that CPM impairs the displacement of the M2-1-P interaction by RNA. Based on the structure of the M2-1-P complex, we generated mutations of the residue Y102 of P and showed that the mutation Y102L of P induces resistance to CPM, similarly to the R151K mutation of M2-1. Together, our results suggest that CPM acts on the M2-1-P interaction, and freezes M2-1 and P dynamics and induces IBs hardening.

## 2. Results

### 2.1. CPM Treatment Induces Hardening of Pseudo-IBs Formed in Cells

We have previously shown that CPM induces IBs hardening in the context of infected cells [15]. Furthermore, using a functional minigenome assay, CPM has been shown to impair hRSV polymerase activity [14]. Here, we decided to study the impact of CPM treatment on pseudo-IBs formed upon the co-expression of hRSV N and P proteins, which are the minimal elements required for LLPS involved in RSV IBs morphogenesis [8]. We first tested the impact of CPM treatment on the fluidity of pseudo-IBs in cells expressing N and P-BFP (P fused to Blue Fluorescent Protein), by performing fluorescence recovery after photobleaching (FRAP) experiments. Compared with the diffusion of P-BFP previously observed in IBs of infected cells [15], P-BFP fluorescence recovery remained below 50% after 1 min, suggesting slower diffusion of P-BFP in N+P pseudo-IBs (Figure 1A). P-BFP fluorescence recovery in pseudo-IBs was not affected by the treatment with 10 µM CPM (Figure 1A). A similar analysis was then performed in the presence of wild type (WT) M2-1 or M2-1_R151K_ mutant. Upon M2-1 expression, the fluorescence recovery of P-BFP in pseudo-IBs was close to 70% after 1 min, similarly to what was observed in hRSV-P-BFP infected cells (Figure 1B), suggesting that M2-1 could play a role in the mobility of P within IBs. A similar fluorescence recovery of P-BFP was obtained in the presence of M2-1_R151K_ (Figure 1C), showing that this M2-1 mutation does not affect the fluidity of pseudo-IBs. Treatment by CPM strongly inhibited P-BFP diffusion in the context of WT M2-1 expression, with only 10% recovery after 1 min, while the treatment had no impact in the presence of M2-1_R151K_ (Figure 1B,C). In parallel, we analyzed the nature of pseudo-IBs by studying their sensivity to hypotonic shock, which disrupts LLPS biocondensates [27,28]. As previously observed for IBs in infected cells [15], in the absence of CPM, pseudo-IBs obtained in the context of co-expression of both N and P disassembled upon a hypotonic shock with or without M2-1 expression, whereas CPM treatment prevented pseudo-IBs disruption only in the presence of M2-1 (Figure 1D).

Altogether, these results highlight the role of M2-1 in the fluidity of hRSV IBs. They also show that the hardening of these IBs by CPM is dependent on the M2-1 protein and reveal that the mechanism of action of CPM is independent of the viral RNA polymerase activity.

### 2.2. Impact of CPM on the Fluidity of Pseudo-IBs Formed In Vitro

We have previously shown that pseudo-IBs form by LLPS upon mixing recombinant fluorescent hRSV N and P proteins in the presence of the crowding agent Ficoll [8,29]. Using this approach, we wanted to assess the impact of CPM on in vitro reconstituted pseudo-IBs in the absence or presence of M2-1. In order to observe M2-1 in these structures, we took advantage of previous data showing that the fusion of mCherry at the C-terminus of M2-1 had no major impact on the RNA polymerase activity [17], and we engineered a recombinant M2-1-mCherry fusion protein expressed in *E. coli*. Both WT and R151K mutated M2-1-mCherry were produced and purified as RNA-free proteins (Figure 2A). An analysis of pseudo-IBs formed in vitro in the presence of recombinant P-BFP and mCherry-N confirmed that the addition of CPM had no major impact on pseudo-IBs morphogenesis (Figure 2B left panel, Appendix A). The addition of M2-1-mCherry to P-BFP and untagged N did not modify pseudo-IBs morphogenesis, and red fluorescence was observed in all of the pseudo-IBs, suggesting that M2-1 is homogeneously incorporated into these structures (Figure 2B right panel, Appendix A).

We then assessed the impact of CPM on P and M2-1 mobility in these in vitro pseudo-IBs by following fluorescence recovery after photobleaching. Here, we used a recombinant P-mCherry protein instead of P-BFP, which was not suited to the FRAP experiment in our conditions. As previously published [8], we first observed that in the presence of only P-mCherry and N, fluorescence recovery of P-mCherry was fast, with 70% recovery after 1 min. In the presence of CPM, P-mCherry was still able to diffuse, although fluorescence recovery was close to 40% after 1 min, suggesting a slight decrease in P mobility in the presence of CPM (Figure 2C). The presence of M2-1 did not affect the mobility of P-mCherry in pseudo-IBs, but incubation with CPM resulted in a strong reduction in P-mCherry mobility, with only 15% recovery after 1 min (Figure 2D). In parallel, we investigated M2-1 diffusion in the context of co-incubation of recombinant N, P, and M2-1-mCherry (Figure 2E). Similar to what we previously observed in infected cells and on reconstituted pseudo-IBs in transfected cells, M2-1 was mobile within in vitro reconstituted pseudo-IBs with a quick recovery of 50% of fluorescence of M2-1-mCherry after 1 min, and the addition of CPM strongly impaired M2-1 mobility (Figure 2E).

These data thus show that CPM induces hardening of in vitro formed pseudo-IBs, revealing that this phenomenon is independent of cellular partners, including RNA. Similar to what was observed in cells for pseudo-IBs, the presence of M2-1 is critical for CPM activity. However, our results also revealed that CPM addition did alter P mobility in the context of N and P co-incubation.

### 2.3. CPM Interferes with the Competitive Binding of P and RNA to M2-1

Based on these results, we hypothesized that CPM may affect the M2-1-P interaction. We have previously shown that the P and the RNA binding sites on the M2-1 core domain partly overlap [23], and that the M2-1-P interaction can be disrupted by tRNA in vitro [19]. We thus studied the impact of CPM on M2-1-P and M2-1-RNA interactions. The capacity of tRNA to compete with P for M2-1-mCherry binding was analyzed using native gel electrophoresis, with M2-1-mCherry migration revealed by UV exposition. As shown in Figure 3A, M2-1-mCherry mobility was strongly modified upon the addition of tRNA or P (lanes 1, 2 and 3), confirming the formation of the corresponding complexes. Of note, the M2-1-mCherry-P complex migrated slightly faster than the M2-1-mCherry-tRNA complex. The addition of CPM did not modify the migration profiles of these complexes (lanes 5, 6, and 7). According to the migration profile, tRNA addition displaced P from the M2-1-mCherry-P complex and led to the formation of the M2-1-tRNA complex (Figure 3A, lanes 3 versus 4). As M2-1 contains two RNA-binding sites, this indicates that tRNA binds at least to the non-specific site that overlaps with the P-binding site. In the presence of CPM, tRNA addition to the M2-1-mCherry-P complex resulted in the formation of a complex that migrated slightly faster than the M2-1-mCherry-P complex (Figure 3A, lanes 7 versus 8). Although this result suggested an inhibition of the switch from M2-1-P to M2-1-tRNA complex in the presence of CPM, the specific migration profile observed could highlight tRNA binding on M2-1 protomers within M2-1-P complex and/or conformational changes of the M2-1-P complex upon the addition of tRNA. In contrast, similar experiments performed in the presence of M2-1_R151K_-mCherry mutant revealed that tRNA was able to displace P from M2-1_R151K_-mCherry in the presence of CPM (Figure 3A, lanes 11–12 and 15–16), validating that the M2-1 R151 residue is critical for CPM activity. To confirm these results and to further analyze the specificity of the CPM effect, we then studied the impact of CPM on the interaction between the P and M2-1 core domain using a pulldown assay. Beads containing GST-M2-1core were incubated with P in the presence or absence of CPM, before the addition of tRNA. After washes, the presence of pulled-down P was analyzed by SDS-PAGE. As shown in Figure 3B, the addition of tRNA induced the release of P in the absence of CPM, but not in the presence of the drug.

Taken together, our observations suggest that CPM interferes with the displacement of P from M2-1 by RNA.

### 2.4. Study of the Potential Interaction of CPM with the M2-1 or M2-1-P Complex

Based on this last result, we considered whether we could evidence a direct effect of CPM on the M2-1-P interaction using Nuclear Magnetic Resonance (NMR). We used the ^15^N-labeled M2-1core domain (~100 µM) and measured the 2D ^1^H-^15^N correlation spectra. CPM did not induce any significant spectral changes in the spectrum of ^15^N-M2-1core (Appendix A), excluding the fact that CPM directly binds to M2-1core. We then probed the binding of a synthetic FITC-P peptide derived from the P-binding motif in M2-1 (FITC-P_95–112_). We followed the titration of ^15^N-M2-1core by FITC-P_95–112_ by 2D NMR. FITC-P_95–112_ induced spectral perturbations (Figure 4A), whereas fluorescein alone did not bind to ^15^N-M2-1core. Saturation was reached with 1–1.5 molar equivalents of peptide. Most perturbed signals (e.g., residue S108 in Figure 4A) displayed an intermediate exchange regime, i.e., broadening beyond detection at mid-titration. This regime is compatible with a Kd in the µM range. Strikingly, perturbed signals delineating the P-binding pocket remained very broad at the titration endpoint (e.g., residues L152 and V156 in Figure 4A). This behavior indicates motions on a µs-ms time scale within the M2-1core-FITC-P_95–112_ complex. CPM did not affect FITC-P_95–112_ binding to WT ^15^N-M2-1core (Appendix A), and did not interact with the peptide, as assessed by the 1D ^1^H NMR spectra (Appendix A).

In order to decipher if CPM could modify the M2-1-P interaction, we also investigated the affinity of FITC-P_95–112_ to the M2-1core domain, in the absence and the presence of CPM using microscale thermophoresis (MST). The FITC-P_95–112_ at 100 nM was incubated in the presence of increasing concentrations of recombinant M2-1core (from 0 to 33.5 µM) alone or in the presence of 1 or 10 µM of CPM. In our conditions the estimated Kd is around 20 µM in the absence and presence of CPM (Appendix A). These results corroborate those obtained by NMR, suggesting that CPM does not affect the affinity between P peptide and M2-1 core domain.

Finally, we assessed if the M2-1 R151K mutation could affect P binding by NMR. This mutation did not alter the fold of the protein, as the 2D ^1^H–^15^N correlation spectra of WT and R151K M2-1core were nearly identical (Appendix A). Titration of ^15^N-M2-1core R151K by FITC-P_95–112_, followed by 2D NMR, displayed similar spectral perturbations to those observed with WT M2-1core (Figure 4C), indicating that complex formation is not impacted by the mutation. However, enhanced signal broadening (e.g., for residues S108 and D155 at 1.5 equivalents of peptide in Figure 4A,C) suggests increased conformational fluctuations within the complex due to the R151K mutation.

In the absence of detection of CPM binding on the M2-1core domain-P peptide or an increase in the affinity of this interaction, possible explanations for these results are that CPM (i) induces a change in the dynamics of the M2-1-P complex, (ii) impairs RNA binding to M2-1 when it interacts with P, or (iii) contributes to the formation of a ternary complex between M2-1, P, and RNA. Of note, the results obtained with the M2-1_R151K_-mCherry mutant suggest that the R151 residue must be involved in competition between P and tRNA for M2-1 binding.

### 2.5. Residue Y102 of P Is Involved in CPM Activity

Based on the results presented above, and the fact that the M2-1_R151K_ substitution induces RSV resistance to CPM [14,15], we wanted to further test the role of P for the activity of CPM. A close inspection of the crystal structure of the M2-1-P complex [24] shows that the M2-1 R151 residue is positioned closely to the P Y102 residue, with both residues having their surface exposed. Their orientations are less well defined than those of residues that are buried in the interface; M2-1 R151K and P Y102 interacting only in one protomer via a non-optimal cation-pi interaction (Figure 5A). We thus wondered if the P Y102 residue could play a role in CPM activity. This residue has previously been shown to be critical for P binding to M2-1, as the Y102A substitution completely abrogated hRSV polymerase activity and P binding to M2-1 [17].

Thus, we first assessed the impact on the polymerase activity of the two P mutations Y102F and Y102L, where tyrosine was replaced by another aromatic and a long hydrophobic residue, respectively, using a functional minigenome assay. As shown in Figure 5B, the P Y102F substitution had no impact on the polymerase activity (*p* > 0.05 using the nested *t* test), whereas Y102L substitution induced a decrease of approximatively 70% of the polymerase activity (*p* < 0.001 using the nested *t* test). Similar experiments performed in the presence of increased concentrations of CPM revealed a dose-dependent inhibition of the polymerase activity, with a similar efficiency when expressing WT P or the Y102F mutant. On the contrary, CPM had no impact on the polymerase activity in the presence of the Y102L P mutant. This result suggests that, although affecting the polymerase activity, this substitution conferred resistance to CPM (Figure 5B). Importantly, as assessed by Western blot, no difference in P expression was observed between WT and P mutants, and CPM did not affect P expression (Figure 5B). Of note, we have previously shown that the M2-1-binding region of the P protein (P residues 97–109) is mainly disordered with a low α-helical propensity in its unbound form in solution [30], but folds into an α-helix when the M2-1-P complex is formed. Mutations inside this region may change the α-helical propensity, and thus the affinity for M2-1. We therefore used the PEP-FOLD 3 server (https://bioserv.rpbs.univ-paris-diderot.fr/services/PEP-FOLD3/ (accessed on 10 January 2023)) to compare the predicted structures of the WT and Y102L mutant P_95–110_ peptides. The predicted α-helical propensity was higher (60%) than what was experimentally observed, but the two peptides did not display a significant difference (Appendix A). This indicates that the mutation would not alter P folding upon M2-1 binding.

We then analyzed the impact of CPM on the capacity of tRNA to compete with the M2-1-P interaction in the presence of CPM, by analyzing the migration of M2-1-mCherry on native agarose gels. The migration of M2-1 complexes with P Y102F and Y102L mutants was similar to that of the WT M2-1-P complex (Figure 5C lanes 3 and 6, compared with Figure 3A), and slightly faster than the M2-1-tRNA complex (Figure 5C, lanes 4 and 7). We thus assessed if tRNA could compete with the M2-1-P interaction in the presence of CPM. As observed for WT P, the presence of CPM impaired the switch from the M2-1-P to the M2-1-tRNA complex for the Y102F mutant of P after the incubation of tRNA. A complex was formed, which migrated faster than M2-1-tRNA and slower than M2-1-P, suggesting that CPM interferes with the competition between tRNA and P for M2-1 binding. On the contrary, in the same conditions, tRNA was able to displace the Y102L mutant of P from M2-1, as a shift corresponding to the formation of the M2-1-tRNA complex was observed. Overall, our results show that similarly to the residue R151 of M2-1, the residue Y102 of P is critical for CPM activity. Thus, although we did not manage to detect a signicant impact of CPM on M2-1-P affinity these data suggest that CPM may act on the M2-1-P complex.

## 3. Discussion

RSV viral factories, as well the protein-protein interactions sustaining their activity, represent original targets for the development of new antiviral strategies. We recently showed that CPM inhibits RSV replication by hardening IBs [15]. However, although the R151K M2-1 mutation has been shown to induce resistance to CPM treatment, the exact molecular mechanism of CPM action remains unknown. Here, by studying pseudo-IB properties in cells or in vitro using FRAP, we first demonstrated that the activity of CPM on IBs relies on the presence of M2-1, and may also impact P mobility in vitro, independently of the polymerase activity or the presence of cellular proteins. Our results also revealed that within pseudo-IBs formed in cells, the presence of M2-1 increased P mobility compared with the condition where only N and P were expressed. On the contrary, in a minimal system of in vitro reconstituted pseudo-IBs, the addition of M2-1 did not affect P mobility. These observations suggest that transient interactions between P and its partners could be a key factor of IB fluidity and dynamics.

Using in vitro approaches, we then showed that CPM impairs the formation of an M2-1-tRNA complex, when M2-1 is already in complex with P. This observation could explain previous results showing that M2-1 presents a diffuse localization within IBs upon CPM treatment of infected cells, contrasting with its concentration into IBAGs together with viral mRNAs, as observed in untreated cells [15]. Based on these results, we hypothesized that CPM could bind to M2-1-P complex. We thus thought to identify P residues that could interact with CPM. An analysis of the crystal structure of M2-1-P complex revealed that within the heterotetramer, the interaction between M2-1 and P slightly differed, in particular the orientation of the lateral chains of the residues R151 of M2-1 and Y102 of P. By investigating the impact of Y102F and Y102L substitutions on P, we observed that the P Y102L substitution induced a strong decrease in the polymerase activity, but also viral resistance to CPM treatment. This result reveals the implication of this P residue in CPM activity. However, we were not able to validate the direct interaction of CPM with the M2-1-P complex using MST or NMR approaches with the M2-1core domain and P peptide. We hypothesized that it could be explained by the poor water solubility of CPM, and a potential low affinity of CPM for the complex.

Our results also suggest that resistance to CPM does not depend on conformational changes of the M2-1 and P binding sites. The direct implication of residues R151 of M2-1 and Y102 of P in a ternary complex with CPM thus remains to be determined. Interestingly, we recently showed that CPM also inhibited bRSV replication [16]. Although the structure of the bRSV M2-1-P complex is not available, the sequence alignment of the domains of M2-1 and P involved in the interaction shows that residues R151 of M2-1 and Y102 of P are conserved among orthopneumoviruses. During the course of this study, we probed the binding of CPM on the M2-1-P complex by crystallography. However, we did not succeed to observe CPM in crystals of M2-1 in complex with P peptide. To receive additional insight, we thus performed docking experiments using the M2-1 tetramer, alone or in complex with a P_90-110_ peptide (PDB accession code 6g0y). Many poses with M2-1 alone show CPM docking in the P-binding groove of M2-1, in the vicinity of R151 (Figure 6A). Some poses show CPM on the other side of the disk, close to where RNA was shown to bind in the crystal structure of the complex [25]. Others show binding on the edge of the disk. With the M2-1-P heterotetramer, CPM predominantly docked next to the “specific RNA binding site” and on the edges of the disk formed by the tetramer (Figure 6B). Overall, this indicates that CPM might explore different binding sites, including the binding sites of P and RNA, and thus interfere with M2-1-RNA and M2-1-P complex formation.

In conclusion, further structural characterization of the mechanism of CPM action on the RSV M2-1-P complex would be necessary in order to allow for a potential rational optimization of this antiviral compound. However, this study clearly suggests that the stabilization of the protein-protein interaction is a potent new approach to specifically modify LLPS dynamics and function. Given the central role of viral factories formed by LLPS upon MNV infections and the numerous transient protein-protein interactions involved in their functioning, our data open new perspective to develop specific inhibitors against all these viruses.

## 4. Materials and Methods

### 4.1. Plasmid Constructs

All of the viral sequences were derived from the hRSV strain Long, ATCC VR-26 (Genbank accession n° AY911262.1). Expression plasmids pCI-N, pCI-P, and pCI-M2-1 were obtained by cloning mammalian codon optimized coding sequences of N, P, and M2-1 into pCI (Genbank accession n° U47119). The coding sequences were amplified by PCR (Phusion High-Fidelity DNA Polymerase, Thermofisher, Waltham, MA, USA) using specific primers (sequences available upon request) and were cloned into pCI by standard molecular biology procedures using NheI and EcoRI for N, MluI and XbaI for P, and MluI and NotI for M2-1. The BFP coding sequence was inserted into the mammalian codon optimized coding sequence of P between residues 76 and 77 as described in (ref risso ballester 2021) to obtain the pCI-P-BFP expression vector. Plasmids for minigenome assay expressing the hRSV N, P, M2-1, and L proteins designated pN, pP, pM2-1 and pL, as well as the pM/Luc subgenomic minigenome expressing the firefly luciferase (Luc) reporter gene under the control of the M/SH gene start sequence were described previously [19].

For the bacterial expression of recombinant mCherry-N, N, P-BFP, P, and M2-1 core proteins, previously described pET-mCherry-N, pET-N, pGEX-P-BFP, pGEX-P, and pGEX-M2-1core plasmids were used [8,23,31]. For expression of recombinant M2-1-mCherry fusion protein, the mCherry gene was amplified by PCR from the pmCherry vector (Clontech, Mountain View, CA, USA) and subcloned at the 3′ end of the M2-1 gene at SmaI-XhoI sites in pGEX-M2-1 plasmid [23]. Point mutations were introduced in pP, pGEX-P, pM2-1, and pGEX-M2-1-mCherry by site-directed mutagenesis, using the Quikchange site-directed mutagenesis kit (Stratagene, San Diego, CA, USA). A sequence analysis was carried out to check the integrity of all of the constructs.

### 4.2. Cells

BSRT7/5 cells (BHK-21 cells that constitutively express the T7 RNA polymerase2) [32] and HEp-2 cells (ATCC: CCL-23) were maintained in DMEM and MEM supplemented with 10% heat-inactivated fetal bovine serum (FBS) and glutamine penicillin–streptomycin solution, respectively. Cells were grown in an incubator at 37 °C in 5% CO_2_. Transfection were performed with 2.5 µL of Lipofectamine 2000 (Thermofisher) per 1 µg of DNA according to the manufacturer’s instructions.

### 4.3. Time-Lapse Microscopy and Photobleaching Experiments on Pseudo-IBs

Live-cell imaging and FRAP experiments were realized using HEp-2 cells seeded in 15 or 4-well Ibidi µ-Slide dishes with a polymer coverslip bottom and transfected with 0.22 µg pCI-P-BFP, 0.22 µg of pCI-N, and 0.056 µg of pCI-M2-1 (WT or R151K mutant) for 24 h. For the FRAP experiments, imaging was performed using Leica SP8 inverted scanning confocal microscope with a 63× oil-immersion objective and a ×8 numerical zoom. Cells were maintained in a climate-controlled chamber (37  °C, 5% CO_2_) during imaging. FRAP acquisition was performed 1 h after the addition of 10 μM CPM or 0.5% DMSO. FRAP experiments were realized using the following settings: 8 s pre-bleach, 1 ms bleach, and 60 s post-bleach at a frame rate of 1 image every 126 ms. Photobleaching of BFP was performed in a circular region at 100% laser intensity. Post-photobleaching fluorescence signals of the bleached region were quantified using the “ROI Intensity Evolution” tool of the Icy software (version 2.4.2.0) [33], in parallel to regions with identical dimensions in (i) a non-photobleached pseudo-IBs located in the vicinity of the targeted pseudo-IB (controls for potential loss of fluorescence in the imaging field studied) and (ii) a background region. Normalization and averaging of the recovery curves were then performed using the easyFRAP web-based tool [34]. For each experimental condition, two individual experiments were performed (*n* = 2), during which 10 to 12 pseudo-IBs were analyzed. Images from one in cellula FRAP replicate are shown in Figure 1. In vitro pseudo-IBs sizes were quantified using the “Spot detector” tool of the Icy software, incremented with a filter for >90% sphericity to discard artefactual values generated by the detection of two adjacent pseudo-IBs as a single entity.

For osmotic shock experiments, imaging was performed using an Olympus IX73 inverted microscope with a 63× oil-immersion objective and a ×2 numerical zoom. Hypotonic shocks were performed by incubating the cells in 10% MEM diluted in water (*v*/*v*) for 5 min. One image was acquired before hypotonic shock and the same cells were imaged every 1 min for 5 min during the shock. For each experimental data point, the hypotonic shock was applied to an entire well of a 15-well Ibidi µ-Slide dish, and one position was studied in order to keep the cell in focus. Seven to ten acquisitions from two independent experiments were performed. Imaging fields of interest comprised 2 to 4 cells, and IBs present before and 5 min following the hypotonic shock were counted manually.

### 4.4. Photobleaching Experiments on In Vitro Reconstituted Pseudo-Ibs

As previously described [8,29], in vitro pseudo-Ibs were reconstituted in 10–20 µL droplets using mCherry-N and P-BFP proteins (3 and 12 µM, respectively), or N, P-BFP, and mCherry-M2-1 (3, 12, and 12 µM, respectively) recombinant proteins in a 20 mM Tris/HCl pH 8.5, 150 mM NaCl, 10% molecular-crowding agent Ficoll buffer, supplemented with 150 µM CPM or 0.5% DMSO. Drops were incubated on 8-well Ibidi µ-Slide dishes for 30 min before imaging at room temperature, and imaged using a SP8 Leica scanning confocal microscope with a 63× oil-immersion objective and a ×4 numerical zoom.

FRAP experiments were realized using the following settings: 2 s pre-bleach, 1 ms bleach, and 60 s post-bleach at a frame rate of 1 image every 1 or 1.476 s for N/P and N/P/M2-1 experiments, respectively (increase in time interval was applied to overcome additional photobleaching during the post-bleach step of the experiments). Photobleaching of mCherry was performed in a circular region at 100% laser intensity. In Vitro FRAP data processing was identical to the in vitro FRAP data processing described above. Images from one in cellula FRAP replicate are shown in Figure 1.

### 4.5. Minigenome Assay

BSRT7/5 cells at 90% confluence in 96-well dishes were transfected with a plasmid mixture containing 125 ng of pM/Luc, 125 ng of pN, 125 ng of pP, 62.5 ng of pL, and 31 ng of pM2-1, as well as 31 ng of pRSV-β-Gal (Promega, Madison, WI, USA) to normalize transfection efficiencies [19]. Transfections were done in triplicate, and each independent transfection was performed three times. Cells were harvested 24 h post-transfection, then lysed in luciferase lysis buffer (30 mM Tris pH 7.9, 10 mM MgCl_2_, 1 mM DTT, 1% Triton X-100, and 15% glycerol). The luciferase activities were determined for each cell lysate using an Infinite 200 Pro (Tecan, Männedorf, Switzerland) and normalized based on β-galactosidase (β-Gal) expression.

### 4.6. Expression and Purification of the Recombinant Proteins

The *Escherichia coli* BL21 (DE3) bacteria strain (Novagen, Madison, WI, USA) was transformed with the plasmids. Cultures were grown at 37 °C in 2xYT medium containing either 100 µg/mL of ampiciline (pGEX vectors) or 50 μg/mL kanamycin (pET vectors). After 8 h, an equal volume of 2xYT medium containing antibiotic was added to the cultures, and the protein expression was induced through the addition of 80 μg/mL isopropyl β-d-1-thiogalactopyranoside (IPTG) overnight at 28 °C. Bacteria were then harvested by centrifugation. The purification of recombinant N, P, and M2-1core proteins has already been described [8,23,31].

For M2-1-mCherry (WT and R151K mutant) purification, pellets were resuspended in a lysis buffer (20 mM Tris–HCl, pH 7.4, 150 mM NaCl, 0.1% Triton X-100, 1 mg/mL lysozyme, and complete protease inhibitor cocktail (Roche)). After incubation on ice for 1h, the lysates were sonicated and benzonase (Novagen) (final concentration 5 U/mL) was added to the lysate, followed by incubation for 30 min at room temperature, before the addition of NaCl up to a concentration of 1 M. After centrifugation at 10,000× *g* for 30 min at 4 °C, the lysates were incubated with Glutathione-Sepharose 4B beads (GE Healthcare) for 1 h at room temperature. The beads were washed three times with washing high-salt buffer (20 mM Tris–HCl, pH 7.4, 1 M NaCl) and three times with washing low-salt buffer (20 mM Tris–HCl, pH 7.4, 150 mM NaCl). To isolate GST-free M2-1-mCherry, the beads were incubated with thrombin (Novagen). Purified proteins were then loaded on a Hi-Load 16/600 Superdex 200 column (Cytiva) and eluted in 20 mM Tris–HCl, pH 7.4, and 150 mM NaCl. Finally, the proteins were concentrated using a centrifugal concentrator with a MWCO of 100 kDa (Vivaspin turbo 4, Sartorius, Göttingen, Germany).

### 4.7. Band Shift on Native Agarose Gels

Recombinant M2-1-mCherry (3 µM) and P proteins (30 µM) in 20 mM TrisHCl, pH 7.4, and 150 mM NaCl buffer were co-incubated for 1 h at room temperature. CPM (675 µM) or an equivalent volume of DMSO was then added to the samples before incubation for 30 min at room temperature. tRNAs (10 µM) were then added, and the samples were incubated for 1 h at room temperature. Then, 50% sucrose loading buffer was added to the samples before loading on native 0.9% agarose gel stained with SYBR Safe (Invitrogen, Waltham, MA, USA). The migration was performed in Tris–Glycine buffer during 6 h at 80 V, before gel staining with amido black 10B.

### 4.8. Pulldown Assay

GST-M2-1core proteins fixed on beads were incubated in the presence of purified recombinant P protein in PBS, at a final volume of 300 µL, in the presence of 400 µM CPM or DMSO (control condition). After 1 h of incubation under agitation at 4 °C, the beads were rinsed twice in 500 µL PBS and resuspended in the presence of 160 µg/mL of tRNA for 30 min, then washed thrice in 500 µL PBS. Then, 40 µL of Laemmli was added to the beads, before boiling for 5 min at 95 °C for analysis using SDS-PAGE and Coomassie blue staining.

### 4.9. Docking of CPM onto M2-1

The structure of the CPM molecule (CAS 4449-51-8) was retrieved from PUBCHEM and converted into MOL2 format. Docking was performed on the Swiss-dock server (http://www.swissdock.ch/docking# (accessed on 13 January 2023)). Docking was performed on the M2-1 tetramer and on the M2-1-P protomer. Rendering was done with ChimeraX [35].

### 4.10. P Peptide Fold Prediction

Structure predictions were run on the PEP-FOLD 3 server (https://bioserv.rpbs.univ-paris-diderot.fr/services/PEP-FOLD3/ (accessed on 10 January 2023)) [36] using peptides spanning P residues of 95–110. This stretch comprised the M2-1 binding region.

### 4.11. Microscale Thermophoresis

Microscale thermophoresis is a technology that uses the motion of fluorescent molecules along a microscale temperature gradient to detect any changes in their hydration shell, which can be induced by binding to a partner [37,38,39,40]. A fixed concentration of FITC-P_95–112_ (100 nM) was incubated with increasing amounts of M2-1 core domain at room temperature for 15 min in PBS and 1% DMSO containing either 0 or 1 or 10 μM of CPM. The measurements were performed for 30 s using a Monolith NT.115 (NanoTemper Technologies GmbH, Munich, Germany) at 20 °C (blue LED power at 80% and infrared laser power at medium). The data from two independent measurements were averaged and analyzed using the temperature-jump phase and the standard fitting mode (derived from the law of mass action) of the NTAnalysis software (MO.Affinity Analysis v2.2.4, Nanotemper technologies).

### 4.12. NMR

The preparation of ^15^N-labeled M2-1 core domain has been reported before [23]. The same protocol was applied to the wild-type and R151K mutant. The M2-1 core domain was in PBS pH 6.4 buffer supplemented with 1 mM DTT and 7.5% D_2_O to lock the NMR spectrometer frequency. Measurements were carried out on a Bruker 700 MHz spectrometer equipped with a TXO cryoprobe. The temperature was set to 298 K. ^1^H–^15^N correlation spectra were acquired using the BTROSY sequence. NMR data were processed with TopSpin 4.0 software (Bruker) and analyzed with CcpNmr Analysis Assign 3.1 software [41]. ^1^H chemical shifts were referenced to DSS. Chemical shift assignment of the wild-type M2-1 core domain has been reported previously [42].

FITC-P_95–112_ peptide was purchased from Proteogenix. A stock solution was prepared by dissolving the peptide in pure water, and by adjusting the pH to neutral through the addition of 1 M NaOH. The final concentration was 1 mM and determined by UV-VIS, using ε(500 nM) = 80,000 cm^−1^·M^−1^. Cyclopamine (CPM) was dissolved in DMSO-d6 (Eurisotop) at 5 mg/mL, equivalent to 12 mM. Interaction experiments with FITC-P_95–112_ were performed using a constant protein concentration of 105 and 115 µM for WT and R151K mutant, respectively, and by adding small volumes of concentrated peptides, so that the protein dilution remained negligible. ^1^H–^15^N correlation spectra were acquired. Titration points were made with 0, 0.1, 0.25, 0.5, 1.0, and 1.5 molar equivalents of peptide. To test the effect of CPM on the M2-1core-FITC-P_95–112_ complex, 1 molar equivalent of CPM was added to WT ^15^N-M2-1core previously incubated with 0.25 equivalent of FITC-P_95–112_.

Then, 1D ^1^H NMR spectra were acquired for FITC-P_95–112_ (100 µM), CPM (50 µM), and an equimolar mixture of both (100 µM), at 700 MHz ^1^H frequency and at a temperature of 298 K in phosphate saline at pH 6.8.

## Figures and Tables

**Figure 1 ijms-24-13862-f001:**
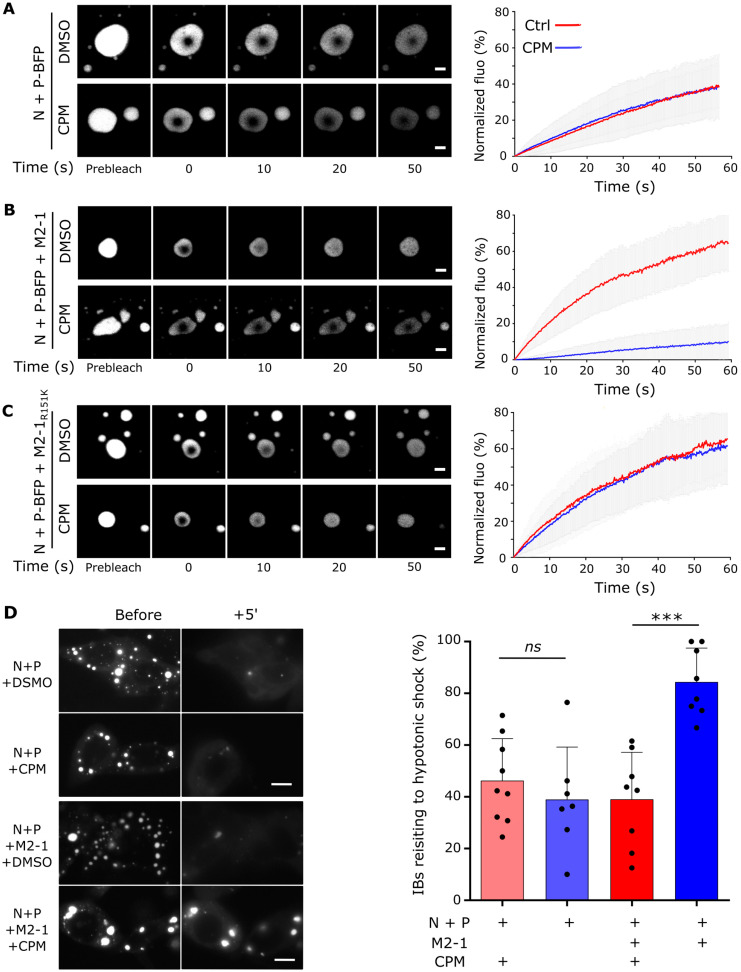
Hardening of pseudo IBs by CPM in cells requires M2-1. (**A**–**C**) P-BFP mobility in biomolecular condensates was analyzed by FRAP in HEp-2 cells transiently expressing N, P-BFP, ±M2-1 (**A**,**B**) or M2-1(R151K) (**C**), and treated with 10 µM CPM (blue) or DMSO (red) for 1 h. (**A**–**C**) Representative images of time-lapse microscopy from FRAP experiments (left panels). Scale bars: 2 µm. Quantification of spontaneous re-distribution of fluorescence after photobleaching, corrected for background and bleaching during post-bleach imaging, and normalized to the post-bleach signals (right panels). Data are represented as mean ± SD of ≥20 FRAP events out of two independent experiments. (**D**) Hypotonic shock was applied on HEp-2 cells transiently expressing N, P-BFP ± M2-1 and treated with 10 µM CPM or DMSO for 1 h. Cells were observed before and 5 min after hypotonic shock (+5’). Representative images of the impact of hypotonic shock on IBs (left panel). Proportions of IBs resisting a 5 min hypotonic shock are represented as mean percentages ± SD in the right panel (8 to 9 acquisitions from two independent experiments). For each condition, the proteins expressed and the presence of CPM are indicated by + below the graph. Significance was tested using an unpaired *t* test followed by Welch correction; *** *p* < 0.001; ns = not significant. Scale bars: 10 µm.

**Figure 2 ijms-24-13862-f002:**
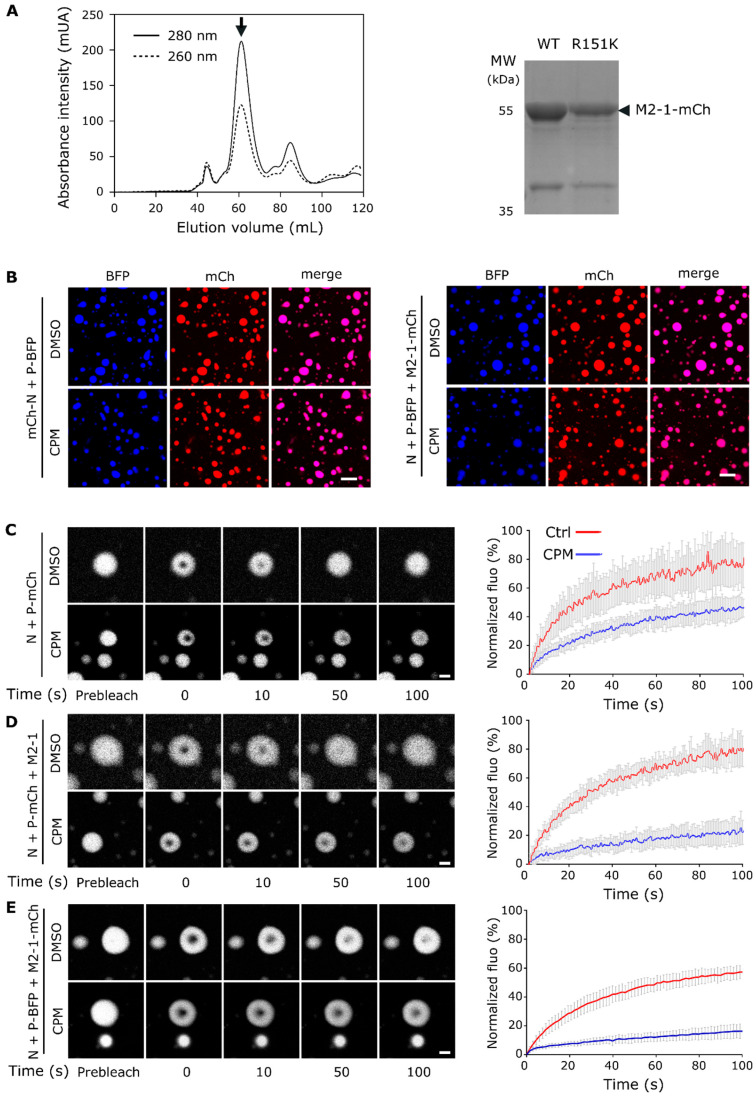
Hardening of in vitro reconstituted pseudo-IBs by CPM depends on M2-1. (**A**) Recombinant GST-M2-1-mCherry proteins (WT and mutant R151K) were produced in *E. coli*. After GST cleavage, gel filtration was performed to isolate M2-1-mCherry (left panel; the arrow shows the peak of elution corresponding to M2-1-mCherry tetramer), and the purified proteins were analyzed by SDS-PAGE and Coomassie blue staining (right panel). (**B**) Recombinant mCherry-N and P-BFP (left panel) or N, P-BFP, and M2-1-mCherry (right panel) proteins were co-incubated and phase separation was assessed using fluorescence microscopy. Scale bars: 10 µm. (**C**–**E**) mCherry-P (**C**,**D**) or M2-1-mCherry (**E**) mobility was analyzed by FRAP in condensates following 30 min of incubation with 150 µM CPM or DMSO. Representative images of time-lapse microscopy from FRAP experiments on in vitro reconstituted pseudo-IBs (left panels). Scale bars: 2 µm. Quantification of the spontaneous re-distribution of fluorescence of mCherry after photobleaching, corrected for background and bleaching during post-bleach imaging and normalized to the post-bleach signal (right panels). Data are represented as mean ± SD of ≥20 FRAP events out of two independent experiments.

**Figure 3 ijms-24-13862-f003:**
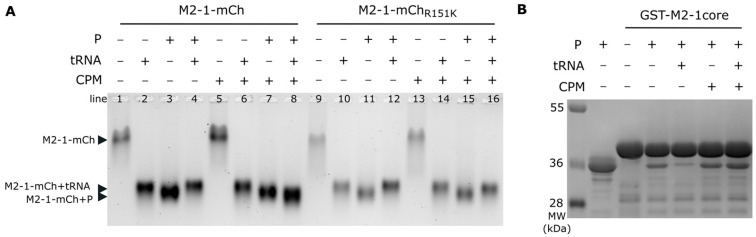
CPM affects competition between P and tRNA for M2-1 binding in vitro. (**A**) Analysis of M2-1-mCherry migration alone or incubated in the presence of tRNA or P, and in the absence or presence of CPM, by native polyacrylamide gel electrophoresis. M2-1-mCherry was observed using UV. (**B**) Pulldown of P by GST-M2-1core in the absence or presence of CPM and/or tRNA, analyzed by SDS-PAGE and Coomassie blue staining.

**Figure 4 ijms-24-13862-f004:**
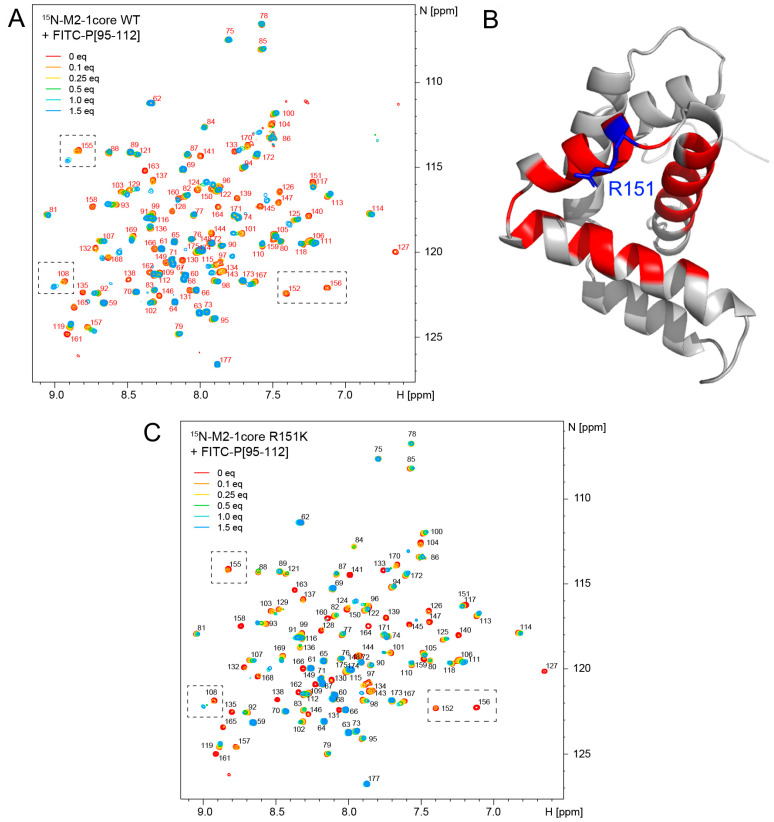
The M2-1-P complex displays motions at the µs-ms time scale. (**A**) Binding of the FITC-P_95–112_ peptide to ^15^N-labeled M2-1core domain was followed by 2D NMR (^1^H frequency 700 MHz, temperature 298 K). Amide signals are annotated with the corresponding residue number. Increasing amounts (0.1 to 1.5 molar equivalents) of FITC-P_95–112_ peptide were added to 100 µM WT M2-1core and a ^1^H-^15^N BTROSY spectrum was acquired at each titrating point. Saturation was reached at a peptide-protein molar ratio of 1.5:1. (**B**) The signals of nearly all residues belonging to the P-binding site remain broad at the last titration point. They are mapped in red on the structure of M2-1core (PDB 2L9J). (**C**) Titration of ^15^N-labeled M2-1core R151K mutant by 0.1 to 1.5 molar equivalents FITC-P_95–112_ peptide is followed by measuring the ^1^H-^15^N BTROSY spectra.

**Figure 5 ijms-24-13862-f005:**
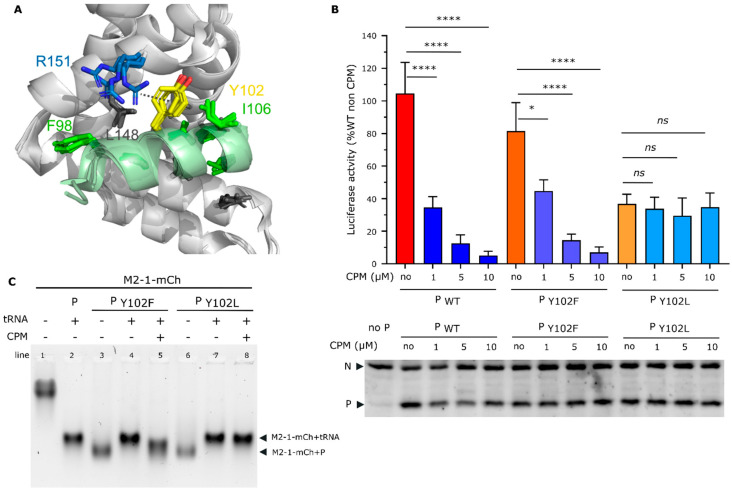
Role of the residue Y102 of P protein in CPM antiviral activity. (**A**) Close-up view of the RSV M2-1-P binding site (PDB 6g0y). The four M2-1 protomers are structurally aligned. The M2-1 core domain is shown by a gray ribbon, and the bound P_90–110_ peptide in green. The side chains of P and M2-1 residues that were previously shown to be critical for RSV replication in vitro by alanine scanning are shown with sticks. The cation-pi interaction between M2-1 R151 (blue) and P Y102 (yellow) is indicated with a broken line. (**B**) Polymerase activity in the presence of mutated P and CPM. BSRT7/5 cells are transfected with plasmids encoding the N, P, L, and M2-1 proteins and the M/Luc subgenomic minireplicon together with pCMV β-gal for transfection standardization. P mutants expressed instead of the corresponding wild-type P are indicated below the histogram. Luciferase activity, reflecting viral RNA synthesis, was measured 24 h after transfection, normalized to the β-galactosidase activity, and expressed as a percentage of the WT protein activity. The mean value ± SD from three independent experiments performed in triplicate or quadruplicate are shown. Nested one-way ANOVA followed by two-sided Dunn’s multiple comparison tests against the untreated group (ns = not significant, **** *p*< 0.0001; * *p* < 0.05). Western blot analysis showing the efficient expression of P mutant proteins in BSRT7/5 cells for one representative experiment is shown. (**C**) Analysis of M2-1-mCherry migration alone or incubated in the presence of tRNA or P mutants, in the absence or presence of CPM, by native polyacrylamide gel electrophoresis. M2-1-mCherry was observed using UV.

**Figure 6 ijms-24-13862-f006:**
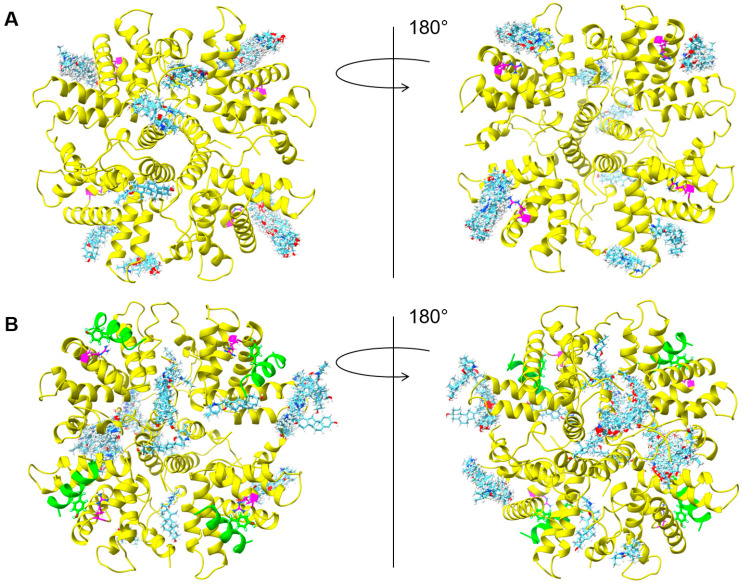
CPM exploring different binding sites on M2-1. Docking experiments with Swiss-dock were performed using (**A**) the M2-1 tetramer without peptide and (**B**) the M2-1 tetramer in combination with the P_95–110_ peptide (PDB 6g0y). Two views, rotated by 180°, and 256 CPM poses are shown for each. M2-1 is the yellow cartoon and the P peptide is the green cartoon. Residue M2-1 R151 is shown with magenta sticks and P Y102 with green sticks, both colored by heteroatoms. CPM is also shown through sticks with carbon atoms in cyan.

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
