# Peer review of "Hardening of Respiratory Syncytial Virus Inclusion Bodies by Cyclopamine Proceeds through Perturbation of the Interactions of the M2-1 Protein with RNA and the P Protein"

_ijms, 2023, doi:10.3390/ijms241813862_

Round 1
Reviewer 1 Report
Respiratory syncytial virus (RSV) RNA syntheses take place in viral factories, which are membrane-less organelles concentrating all the elements required for viral RNA synthesis, (the encapsidated genomic and antigenomic viral RNA, the viral polymerase L and its co-factor P, and the viral transcription factor M2-1).
The authors of the manuscript have recently demonstrated that cyclopamine (CPM) inhibits RSV multiplication by disorganizing and hardening the viral factories.
In this new study, they have investigated the molecular mechanisms of action of CPM. For this, they have used reconstituted systems in cellula (by co-expressing N and P with or without M2-1) and in vitro (using purified recombinant N, P, M2-1 proteins).
They show (i) that CPM activity depends on the presence of M2-1 together with N and P and (ii) that CPM impairs the competition between P and RNA binding to M2-1.
As mutations on both P and M2-1 induced resistance against CPM activity, they suggest that CPM affect the dynamics of the M2-1−P interaction and consequently affect the relative mobility of proteins in the viral factories.
Globally, the experimental data are convincing and support the conclusions of the manuscript.
I have only two concerns:
1) In figure 3, in lane 8, the M2-1-mCH complexed by P migrates slightly faster than in line 7. This is also stated by the authors but they did not provide any explanation.
2) The legend of figure 6 is not enough informative. What is the aromatic amino acid residue in blue in panel 6A? Why are M2-1 R151 and P Y102 not represented in figure 6B ? Why are some CPM represented with cyan sticks and some with cloud of spheres?
Minor remark :
Line 440 and 441 : the authors seem to hesitate between pose and pause.
English seems fine to me
Author Response
I have only two concerns:
1) In figure 3, in lane 8, the M2-1-mCH complexed by P migrates slightly faster than in line 7. This is also stated by the authors but they did not provide any explanation.
We agree with the reviewer that hypotheses concerning the difference of migration should have been mentioned. We modified lines 230-233 : “Although this result suggested an inhibition of the switch from M2-1−P to M2-1−tRNA complex in the presence of CPM, the specific migration profile observed could highlight tRNA binding on M2-1 protomers within M2-1−P complex and/or conformational changes of M2-1−P complex upon tRNA addition.”
2) The legend of figure 6 is not enough informative. What is the aromatic amino acid residue in blue in panel 6A? Why are M2-1 R151 and P Y102 not represented in figure 6B ? Why are some CPM represented with cyan sticks and some with cloud of spheres?
We thank the reviewer to point the lack of clarity of figure 6. The figure 6 and the legend have been revised.
Minor remark :
Line 440 and 441 : the authors seem to hesitate between pose and pause.
We thank the reviewer for this comment. We homogenized the term, “pose”
Reviewer 2 Report
This paper describes a study aimed to clarify mechanisms of antiviral activity of CPM, and many experiments are shown on the effect of CPM over M2-1 complexation. This paper, however, is very difficult to read, and many experiments are fragmentally shown without clear explanations. For examples, complex hardening phenomena of IB occurs in the presence of CPM; CPM partially impairs M2-1−P interactions, which appears to be a complex phenomenon; and NMR studies did not provide meaningful conclusion on the effect of CPM in the protein-protein interactions. How are these results unified? The introduction is not straightforward, and advance made in this work is not concisely summarized. A number of negative data are shown, which make very difficult to understand this work. This referee cannot recommend publication of this work in ijms.
Author Response
This paper describes a study aimed to clarify mechanisms of antiviral activity of CPM, and many experiments are shown on the effect of CPM over M2-1 complexation. This paper, however, is very difficult to read, and many experiments are fragmentally shown without clear explanations. For examples, complex hardening phenomena of IB occurs in the presence of CPM; CPM partially impairs M2-1−P interactions, which appears to be a complex phenomenon; and NMR studies did not provide meaningful conclusion on the effect of CPM in the protein-protein interactions. How are these results unified? The introduction is not straightforward, and advance made in this work is not concisely summarized. A number of negative data are shown, which make very difficult to understand this work. This referee cannot recommend publication of this work in ijms.
Based on reviewer’s comments, we have tried to simplify parts of the manuscript to make it clearer. The introduction is still quite long but we believe that all the information is necessary to understand the complexity of the interactions between M2-1 and P and RNA and the results described in the manuscript.
Regarding the results, we sequentially show in the different paragraphs that:
1/ CPM induces the hardening of pseudo-IBs formed in cells expressing N, P, and M2-1, thus independently of the polymerase activity,
2/ CPM also induces the hardening of pseudo-IBs reconstituted in vitro and that pseudo-IB hardening depends on M2-1, demonstrating that CPM activity is independent of cellular factors.
3/ CPM interferes with the switch from M2-1-P to M2-1-RNA interaction in vitro (CPM does not impair M2-1-P interaction contrary to the reviewer comment),
4/ Despite structural investigation using biophysical approaches, we didn’t manage to identify neither a CPM binding site on M2-1 or M2-1-P complex, nor a potential impact on M2-1-P affinity. Hence our results clearly demonstrate the complexity of the mechanism of action of CPM; and that these results will be useful for further investigation.
5/ We finally identified that mutation Y201L on P induces resistance to CPM treatment, highlighting that both M2-1 and P are involved in CPM activity on IBs.
We regret that the reviewer did not provide any constructive criticism on the quality and interpretation of the experiments performed and the results obtained.
We are convinced that our results are important for the field of viral condensates. It is noteworthy that molecules such as CPM that modify the properties of viral condensates are of most interest not only as new antivirals but also as tools to further understand the dynamics of these organelles.
Round 2
Reviewer 2 Report
This referee does not consider that significant improvements have been made in the revised manuscript. Purpose, explanation, and conclusion are not clear, and this work cannot be recommended for publication in IJMS.
Several other defects.
Section 2.1 and 2.2. That CPM induces the hardening of pseudo-IBs was previously described in reference 15. The present work confirms the previous results.
Without relevant explanations, interactions of CPM are not convincing.
Reply comment 4 is unusual. It is not logical that lack of clear structural results demonstrates complex nature of mechanism.
Author Response
This referee does not consider that significant improvements have been made in the revised manuscript. Purpose, explanation, and conclusion are not clear, and this work cannot be recommended for publication in IJMS.”
As previously answered, we have tried to simplify parts of the manuscript to make it clearer. We believe that all the information is necessary to understand the manuscript, and gave a detailed answer with the first revision. We regret that the reviewer only give a general comment.
Several other defects.
1/ Section 2.1 and 2.2. That CPM induces the hardening of pseudo-IBs was previously described in reference 15. The present work confirms the previous result.
This statement is wrong. The reference 15 refers to the study of CPM activity on IBs from infected cells. Our results are based on pseudo-IBs study, formed in the presence of only N, P and M2-1 viral proteins. This approach is new and clearly shows that CPM activity is independent of viral polymerase activity and of cellular factors.
2/ Without relevant explanations, interactions of CPM are not convincing.
Reply comment 4 is unusual. It is not logical that lack of clear structural results demonstrates complex nature of mechanism.
We suppose that the reviewer still considers that all the biophysical studies performed in order to investigate the interaction of CPM on M2-1 and/or M2-1/P complex, which did not allow to reveal any interaction in our experimental conditions, are not informative. We regret that the reviewer has maintained this position following our response, but we consider that these are important elements to know for any future study. Indeed, CPM is a novel antiviral of interest whose optimization could pave the way for new therapeutic molecules. Our results show that, in the absence of a clear mechanism of action, the rational optimization of this molecule appears complex.